# NH_4_^+^ Toxicity, Which Is Mainly Determined by the High NH_4_^+^/K^+^ Ratio, Is Alleviated by CIPK23 in *Arabidopsis*

**DOI:** 10.3390/plants9040501

**Published:** 2020-04-14

**Authors:** Sujuan Shi, Fangzheng Xu, Yuqian Ge, Jingjing Mao, Lulu An, Shuaijun Deng, Zia Ullah, Xuefeng Yuan, Guanshan Liu, Haobao Liu, Qian Wang

**Affiliations:** 1Tobacco Research Institute, Chinese Academy of Agricultural Sciences, Qingdao 266101, China; shisujuan2014@163.com (S.S.);; 2Graduate School of Chinese Academy of Agricultural Sciences, Beijing 100081, China; 3Shandong Province Key Laboratory of Agricultural Microbiology, Shandong Agricultural University, Tai’an 271017, China

**Keywords:** NH_4_^+^ toxicity, *CIPK*, K^+^, leaf chlorosis, root development

## Abstract

Ammonium (NH_4_^+^) toxicity is always accompanied by ion imbalances, and NH_4_^+^ and potassium (K^+^) exhibit a competitive correlation in their uptake and transport processes. In *Arabidopsis thaliana*, the typical leaf chlorosis phenotype in the knockout mutant of *calcineurin B-like interacting protein kinase 23* (*CIPK23*) is high-NH_4_^+^-dependent under low-K^+^ condition. However, the correlation of K^+^ and NH_4_^+^ in the occurrence of leaf chlorosis in the *cipk23* mutant has not been deeply elucidated. Here, a modified hydroponic experimental system with different gradients of NH_4_^+^ and K^+^ was applied. Comparative treatments showed that NH_4_^+^ toxicity, which is triggered mainly by the high ratio of NH_4_^+^ to K^+^ (NH_4_^+^/K^+^ ≥ 10:1 for *cipk23*) but not by the absolute concentrations of the ions, results in leaf chlorosis. Under high NH_4_^+^/K^+^ ratios, *CIPK23* is upregulated abundantly in leaves and roots, which efficiently reduces the leaf chlorosis by regulating the contents of NH_4_^+^ and K^+^ in plant shoots, while promoting the elongation of primary and lateral roots. Physiological data were obtained to further confirm the role *CIPK23* in alleviating NH_4_^+^ toxicity. Taken all together, *CIPK23* might function in different tissues to reduce stress-induced NH_4_^+^ toxicity associated with high NH_4_^+^/K^+^ ratios by regulating the NH_4_^+^–K^+^ balance in *Arabidopsis*.

## 1. Introduction

Appropriate nitrogen (N) and potassium (K) fertilizers are important for plant development and growth [1,2]. As a paradoxical inorganic compound, ammonium (NH_4_^+^) is a main type of N fertilizer for plants, but its high accumulation in plant cells is strongly toxic [1,3,4]. In recent years, due to the abuse and low utilization of N fertilizers, crops have widely suffered from NH_4_^+^ toxicity triggered by excessive NH_4_^+^ (2–20 mM), especially in irrigated paddy fields [2,5,6,7]. Meanwhile, K^+^ deficiency has been reported to be a common problem in more regions [3]. According to statistics, ~70% of the rice paddies in southeastern China and ~60% of the wheat belts in southern Australia are facing K^+^ deficiency [8,9]. There is a competitive correlation between the uptake of NH_4_^+^ and K^+^ due to their similar charge, size and hydration energy characteristics [10,11]. The toxicity caused by high external NH_4_^+^ concentrations is greatly aggravated by K^+^ deficiency, thereby resulting in more serious yield reduction [1,3]. Optimizing the proportions of NH_4_^+^ and K^+^ in soils is essential for reducing NH_4_^+^ toxicity and boosting crop yield.

Many investigations have been carried out to explain the interaction of K^+^ and NH_4_^+^ in plants; however, only a limited number of genes have been identified to be involved in the uptake and transport of both ions. Among them, *Arabidopsis thaliana* calcineurin B-like (CBL) interacting protein kinase 23 (CIPK23) is considered to be a notable regulator in the acquisition of K^+^ and NH_4_^+^, as well as NO_3_^−^, Mg^2+^ and other ions [12,13,14,15,16,17,18]. CIPK23 was first identified to activate a Shaker channel Arabidopsis K^+^ transporter 1 (AKT1) under low-K^+^ stress [17]. The kinase is recruited to the plasma membrane of root cells by CBL1/9 to form the CBL1/9-CIPK23 complex, which greatly enhances the activity of AKT1 by phosphorylation and subsequently promotes K^+^ uptake by roots under low-K^+^ (0.1 mM) stress [17,18]. *CIPK23* knockout mutant *cipk23* shows a typical leaf chlorosis phenotype and a faster root elongation in low-K^+^ medium [17]. Another research group found that the CBL1-CIPK23 complex regulates high-affinity K^+^ transporter 5 (HAK5)-mediated high-affinity K^+^ uptake in *Arabidopsis* roots [15]. However, in this study, a remarkable reduction in plant growth (without leaf chlorosis symptoms) was observed in *cipk23* under low-K^+^ (0.01, 0.1 and 0.5 mM) treatments [15].

As phenotype identification is a key step in stress treatments, a detailed comparison between the two experimental systems was then conducted. In the study by Xu et al. [17], K^+^ in the Murashige and Skoog (MS) medium was replaced with NH_4_^+^ for the purpose of forming a low-K^+^ stress. As a matter of fact, a low-K^+^ (0.1 mM) but high-NH_4_^+^ (30 mM) plating medium system was generated. Ragel et al. [15] used a low-K^+^ (0.01, 0.1 and 0.5 mM) culture solution with no NH_4_^+^ in a hydroponic growing system. Thus, high levels of NH_4_^+^ might be the indispensable factor in leaf chlorosis. This is also consistent with a previous report stating that leaf chlorosis under low-K^+^ conditions is high-NH_4_^+^-dependent, although this observation was not further explained [17]. Coincidently, a recent study reported that CIPK23 was found to restrain NH_4_^+^ transport by inhibiting the activity of ammonium transporters (AMT1, AMT1;1 and AMT1;2) via phosphorylation [13]. It is noteworthy that a hydroponic system with much higher K^+^ concentrations (1 or 5 mM) and medium NH_4_^+^ concentrations (1 or 2 mM) was used in this work; no chlorosis phenotype but less root and shoot biomass was observed in the *cipk23* mutant [13]. Based on the above analysis of the relationship between the occurrence of symptoms and the concentrations of ions, the following questions were raised: What is the essence of leaf chlorosis in *cipk23*? Is the phenotype actually triggered by high concentrations of NH_4_^+^, low concentrations of K^+^ or the ratio of NH_4_^+^ to K^+^ (NH_4_^+^/K^+^)? How does CIPK23 function in the inhibition of leaf chlorosis?

To clarify these questions, a hydroponic experimental system using nitrate (NO_3_^−^) as the sole basic nitrogen source was used. Different gradients of NH_4_^+^ and K^+^ were applied to form solutions with different NH_4_^+^/K^+^ ratios. We found that leaf chlorosis of *cipk23* mutant seedlings is triggered by NH_4_^+^ toxicity. The relatively higher NH_4_^+^/K^+^ ratio in leaves, but not the respective concentration of either ion, results in NH_4_^+^ toxicity. *CIPK23* greatly enhances tolerance to NH_4_^+^ toxicity in plants by regulating the NH_4_^+^ and K^+^ contents in leaves and reduces the light-induced damage and the generation of reactive oxygen species (ROS) in leaves. It also promotes root development under stresses associated with high NH_4_^+^/K^+^ ratios. The gene might participate in the plant tolerance to NH_4_^+^ toxicity triggered by high NH_4_^+^/K^+^ ratios and act as a balancer in leaf NH_4_^+^-K^+^ homeostasis.

## 2. Results

### 2.1. CIPK23 Alleviates Leaf Chlorosis Caused by Excessive NH_4_^+^ Accumulation in Arabidopsis

One distinction between NH_4_^+^ and K^+^ is that NH_4_^+^ undergoes many chemical reactions, while K^+^ remains in an ionic state [1]. Although previous studies have shown that the leaf chlorotic phenotype of the *cipk23* mutant is low-K^+^- and high-NH_4_^+^-dependent [17], the changes in the ion content in *cipk23* and wild-type Col-0 under different conditions have not been determined, which might provide us some clues. The same *cipk23* T-DNA insertion mutant (SALK_036154), which was widely used in the previous studies of K^+^ and NH_4_^+^ uptake, was selected. The mutant is verified to be a mutant null, and the T-DNA insertion results in a complete loss of *CIPK23* transcript (Appendix A). This mutant allele was found to exhibit a severe leaf chlorosis phenotype under low-K^+^–high-NH_4_^+^ stress [17]. When complemented with the *CIPK23* cDNA, the complemented mutant lines were no longer sensitive to the stress [17]. Meanwhile, the mutant also showed an obvious growth inhibition phenotype to high-NH_4_^+^ stress [13]. Here, four treatments with different NH_4_^+^ and K^+^ supplies were analyzed with a plating medium. ([NH_4_^+^]-[K^+^]) is used to represent the respective concentrations of NH_4_^+^ and K^+^ (mM) in each treatment.

Under high-NH_4_^+^–high-K^+^ (MS medium, 20.6-20.25) and no-NH_4_^+^–low-K^+^ (0-0.1) conditions, no significant phenotypic differences were observed between Col-0 and *cipk23* mutant plants (Figure 1a). All seedlings showed slight yellowing of their cotyledons under no-NH_4_^+^–low-K^+^ (0-0.1) condition, compared to those growing on MS plates (Figure 1a). Under the high-NH_4_^+^–low-K^+^ (30-0.1) treatment, both materials showed an obvious suppression in root elongation, while typical leaf chlorosis and fewer lateral roots were observed on *cipk23* mutant plants (Figure 1a). When the NH_4_^+^ concentration was decreased to 10 mM with 0.1 mM K^+^ (10-0.1), reduced stress effects were observed on the roots of Col-0 and the *cipk23* mutant, compared with the high-NH_4_^+^–low-K^+^ (30-0.1) treatment. Only the *cipk23* mutant exhibited chlorosis of the cotyledons (Figure 1a). Based on the phenotypic analysis under the four treatments, it was clear that leaf chlorosis was not triggered by the high-NH_4_^+^ condition, for which *cipk23* seedlings did not show the sensitive phenotype on MS plate with a high NH_4_^+^ concentration (20.6 mM), or by the low-K^+^ condition, for which *cipk23* also did not show the phenotype on no-NH_4_^+^–low-K^+^ (0-0.1) medium with a low K^+^ concentration (0.1 mM). The results indicate that the leaf chlorosis and growth inhibition in the *cipk23* mutant might be related to higher NH_4_^+^/K^+^ ratios.

Ion content analysis was then conducted. It was found that the NH_4_^+^ content in the shoots and roots of the *cipk23* plants was greatly enhanced in two high-NH_4_^+^ treatments (Figure 1b). Although the MS medium contained a high concentration of NH_4_^+^, a much lower NH_4_^+^ accumulation was detected, suggesting a high assimilation efficiency. As predicted, the K^+^ content of the *cipk23* mutant was significantly decreased under all treatments as compared to Col-0 (Figure 1c). As leaf chlorosis and root growth are typical syndromes of NH_4_^+^ toxicity, the ion content analysis indicated that the chlorotic phenotype of the *cipk23* mutant might result from the NH_4_^+^ toxicity triggered by excessive NH_4_^+^ accumulation. The phenotype test and ion content analysis suggested that the leaf chlorosis and growth inhibition in the *cipk23* mutant might be induced by high NH_4_^+^/K^+^ ratio triggering NH_4_^+^ toxicity.

To confirm the effect of NH_4_^+^ toxicity in the *cipk23* mutant, the hydroponic growth system used by Ragel et al. [15] was applied. NO_3_^-^ is the main nitrogen source in this system. Typically, NH_4_^+^ toxicity always occurs when the external NH_4_^+^ concentration in the cultivation environment is 5 mM or higher (6 and 10 mM in most studies) [11,19,20,21]. Here, 2 mM NH_4_^+^ with 0.1 mM K^+^ was used to generate a high NH_4_^+^/K^+^ ratio (20:1). All the seedlings grew hydroponically in 1/5 Hoagland Solution (0-1) for 15 days and were then transferred to different NH_4_^+^/K^+^ ratio conditions and treated for 7 days.

Phenotypic analysis indicated that, under no-NH_4_^+^–high-K^+^ (0–1) and no-NH_4_^+^–low-K^+^ (0-0.1) conditions, no leaf chlorosis and a slight biomass reduction were observed in *cipk23* mutant as compared to Col-0 (Figure 2a). However, when the mutant grew under high-NH_4_^+^–low-K^+^ (2-0.1) treatment, the typical leaf chlorosis with biomass reduction emerged (Figure 2a). Ion content analysis showed that under the high-NH_4_^+^–low-K^+^ (2-0.1) treatment, compared to Col-0, the NH_4_^+^ content of both the shoots and roots of the *cipk23* mutant was greatly enhanced by ~133% and 90%, respectively (Figure 2c). No obvious change in NH_4_^+^ content was observed in the other two treatments. The *cipk23* mutant exhibited a lower K^+^ content under all three treatments (Figure 2c). The transcription level of *CIPK23* in Col-0 leaves under the three treatments was also investigated by RT-qPCR (Figure 2b). The transcription of *CIPK23* was increased slightly under no-NH_4_^+^–low-K^+^ (0-0.1) condition and was evidently upregulated by high-NH_4_^+^–low-K^+^ (2-0.1) stress (Figure 2b).

Phenotypic analysis and ion content measurement of the two different systems indicated that excessive NH_4_^+^ accumulation led to leaf chlorosis in the *cipk23* mutant. The nature of leaf chlorosis is NH_4_^+^ toxicity, which is probably triggered by high NH_4_^+^/K^+^ ratios. *CIPK23* alleviates the leaf chlorosis and biomass reduction caused by NH_4_^+^ toxicity. In addition, *CIPK23* plays a general and facilitating role in K^+^ absorption in *Arabidopsis*.

### 2.2. High NH_4_^+^/K^+^ Ratio is the Key Determinant Triggering NH_4_^+^ Toxicity

NH_4_^+^ toxicity leads to severe leaf chlorosis and a decline in the biomass of plants. The toxicity is closely linked to the K^+^ concentration, and various experiments in other plants have shown that it can be ameliorated by the addition of K^+^ and aggravated by K^+^ deficiency [10,21]. In view of the above analysis, a high NH_4_^+^/K^+^ ratio is probably the main cause of NH_4_^+^ toxicity.

To verify this hypothesis, a variety of relatively higher NH_4_^+^/K^+^ ratio conditions were established using the hydroponic system (Figure 3). Under these stresses, 15-day-old seedlings were treated for 7 days. It was found that, when the NH_4_^+^/K^+^ ratio is 10:1 or 20:1, the *cipk23* mutant showed typical chlorosis on the bottom leaves and evident growth reduction as compared to Col-0. When the NH_4_^+^/K^+^ ratio was reduced to 4:1 (lower than 10:1), no leaf chlorosis but only growth inhibition was observed in the *cipk23* mutant. When the NH_4_^+^/K^+^ ratio was increased to 50:1, both Col-0 and the *cipk23* mutant showed obvious toxicity symptoms, and the chlorosis and biomass reduction of the mutant were more serious (Figure 3a,b). The phenotypic analysis was quite consistent with the following chlorophyll content and fresh weight measurements (Figure 3c,d).

To further confirm the relationship between high NH_4_^+^/K^+^ ratios and the occurrence of chlorosis, the phenotypes of the *cipk23* mutant and Col-0 and their chlorophyll content and fresh weight under low NH_4_^+^/K^+^ ratio (ratio = 1:1) conditions were detected (Figure 4). Here, the low NH_4_^+^/K^+^ ratio conditions were divided into two groups: low-NH_4_^+^–low-K^+^ and high-NH_4_^+^–high-K^+^ conditions (Figure 4a,b). In these conditions, all the plants grew well and remained green. No evident differences were observed in the leaves, chlorophyll content and biomass accumulation between *cipk23* mutant and Col-0 (Figure 4c,d). The results showed that the lack of *CIPK23* had little effect on plant growth under low NH_4_^+^/K^+^ ratio conditions.

Taken together, the series of trials indicated that the occurrence of chlorosis is not related to the absolute concentration of NH_4_^+^ (from 1 to 50 mM) or K^+^ (from 0.05 to 1 mM); rather, it is determined only by high NH_4_^+^/K^+^ ratios. In other words, a high NH_4_^+^/K^+^ ratio is the necessary prerequisite of NH_4_^+^ toxicity. *CIPK23* greatly enhances the tolerance of *Arabidopsis* to NH_4_^+^ toxicity caused by high NH_4_^+^/K^+^ ratio (ratio ≥ 10:1) stresses.

### 2.3. The Transcriptional Level of CIPK23 in Leaves Was Upregulated by High-NH_4_^+^ Conditions

In terms of the different performances of *cipk23* and Col-0 under high and low NH_4_^+^/K^+^ ratio conditions, it is necessary to investigate the expression pattern of *CIPK23*. GUS staining was then performed using the *ProCIPK23*::*GUS* material. Fifteen-day-old seedlings cultured under no-NH_4_^+^–high-K^+^ (0-1) condition were transplanted into different conditions for 7 days and then collected for GUS straining. Under no-NH_4_^+^–high-K^+^ (0-1) condition, *CIPK23* was expressed mainly in the petioles of leaves at a very low level (Figure 5a). When the seedlings were treated under high NH_4_^+^/K^+^ ratio conditions, enhanced GUS activity was detected in the blades, veins and petioles of the leaves. The expression pattern of *CIPK23* under most low NH_4_^+^/K^+^ ratio conditions was at a low level, similar to that under no-NH_4_^+^–high-K^+^ (0-1) condition, with an exception for the 10-10 condition (Figure 5b). In conclusion, the expression of *CIPK23* was significantly upregulated not only under high NH_4_^+^/K^+^ stresses but also at high NH_4_^+^ levels.

### 2.4. CIPK23 Promotes Root Growth by Enhancing the Elongation of Primary and Lateral Roots

In plants, the plasticity of the root architecture is crucial for the efficient acquisition of mineral nutrients from soils [22]. Meanwhile, root growth inhibition is also a typical syndrome of NH_4_^+^ toxicity [2]. Although the gene was found to be involved in the balance of K^+^ and NH_4_^+^ (Figure 1, Figure 2, Appendix A and Figure 6i,j), its function in root growth and development is largely unknown. To better understand the influence of *CIPK23* on root growth, medium-NH_4_^+^–medium-K^+^ ratio (0.5-0.5) and high-NH_4_^+^–medium-K^+^ ratio (10-0.5) conditions were selected as the typical low NH_4_^+^/K^+^ ratio and high NH_4_^+^/K^+^ ratio treatments, respectively, and applied in the subsequent experiments. The two conditions were selected based on the general performance of Col-0 and the *cipk23* mutant (Figure 6).

The expression of *CIPK23* in root tissues under 0.5-0.5 and 10-0.5 conditions was determined by RT-qPCR (Figure 6a). In roots, *CIPK23* was expressed at a high level under the low NH_4_^+^/K^+^ ratio condition and showed a slight increase under the high NH_4_^+^/K^+^ ratio condition for 7 days, indicating that *CIPK23* might be expressed constitutively in roots and its expression might not be closely related to NH_4_^+^/K^+^ ratio stresses. The result was consistent with that from the GUS staining assay (Figure 6b and Appendix A).

Root analysis indicated that under the 0.5–0.5 condition or other low NH_4_^+^/K^+^ ratio conditions, the *cipk23* plants displayed a general deficiency in fresh weight, biomass accumulation and the length of roots (primary roots, first and secondary lateral roots), with a higher cellular NH_4_^+^/K^+^ ratio (Figure 6 and Appendix A). High NH_4_^+^/K^+^ ratio conditions intensified the difference between the *cipk23* mutant and Col-0. It can be easily observed that the length of the primary roots and first lateral roots of the *cipk23* mutant were significantly shorter than those of Col-0 under 10-0.5 condition (Figure 6e,f). Elongation of the secondary lateral root of the *cipk23* mutant was severely inhibited under the 10-0.5 condition (Figure 6g). The results indicated that *CIPK23* effectively facilitated root growth and development, especially in the process of root elongation.

### 2.5. The Tolerance to Light-Induced Damage is Decreased and the Generation of ROS is Increased in cipk23 Mutant under High NH_4_^+^/K^+^ Stress

As shown in Figure 7a,b, the absence of *CIPK23* in *Arabidopsis* greatly led to declines of growth and dry weight, especially in high NH_4_^+^/K^+^ ratio (10-0.5) condition. It is postulated that the gene might be involved in photosynthesis [23,24]. Thus, some chlorophyll fluorescence characteristics of Col-0 and *cipk23* under the two selected conditions, including *F_v_*/*F_m_*, non-photochemical quenching (NPQ, qN) and the fluorescence decrease ratio (Rfd), were then evaluated by using kinetic chlorophyll fluorescence imaging systems (FluorCam, PSI) [23,25,26,27,28].

Under low NH_4_^+^/K^+^ ratio conditions, no substantial differences in these parameters were detected between the two plant materials. Under high NH_4_^+^/K^+^ ratio conditions, the NPQ and Rfd values of Col-0 were significantly higher than those of the *cipk23* mutant (Figure 7c–e). NPQ is an indicator of excessive light and heat dissipation in the PSII system under illumination conditions, and Rfd can reflect the tolerance of plants under stress. The measurement results suggested that *CIPK23* might improve plant tolerance to high NH_4_^+^/K^+^ ratios by avoiding light-induced damage.

Excessive amounts of ROS are generated in plants when they suffer from abiotic and biotic stresses [29]. Meanwhile, NH_4_^+^ increases mitochondrial ROS production [30]. The primary ROS include superoxide, hydrogen peroxide (H_2_O_2_) and nitric oxide [31]. The H_2_O_2_ content of leaves was detected by DAB staining. Under low NH_4_^+^/K^+^ ratio (0.5-0.5) condition, a small amount of H_2_O_2_ was produced in Col-0 and *cipk23* leaves (Figure 7f). Under high NH_4_^+^/K^+^ ratio conditions, a large amount of H_2_O_2_ was generated in the bottom leaves of the *cipk23* mutant (Figure 7f), indicating that *CIPK23* might be helpful for plants to establish an efficient self-protection mechanism and reduce ROS-triggered damage.

### 2.6. Knockout of CIPK23 Results in Chlorophyll Degradation under High NH_4_^+^/K^+^ Stress

The toxic effects of ammonium to plants have been explained by several mechanisms: damaged chloroplast ultrastructure, plant senescence and the disruption of photosynthesis [2,19]. Some marker genes are used widely in these processes. Among them, *NAC* (*NAM*, *ATAF* and *CUC*) transcription factor *ORE1*/*NAC2* is used as a positive gene in leaf senescence, *STAYGREEN1* (*SGR1* or *NYE1*) is reported to be a critical gene in chlorophyll degradation and *WRKY70* is identified as a negative regulator in plant developmental senescence [32,33,34,35,36,37]. To determine whether leaf chlorosis at high NH_4_^+^/K^+^ ratios in this study triggers these regulatory mechanisms, the expression of some indicative genes was detected by RT-qPCR. The 15-day-old seedlings were treated under two conditions, and leaf samples (the 5th and 6th leaves from the bottom up) of Col-0 and the *cipk23* mutant were collected on the 7th and 16th days for RT-qPCR detection (Figure 8a,c). It should be noted that the fifth and sixth leaves were the newly grown leaves after the seedlings were transferred. Quantitative analyses showed that, under 0.5-0.5 condition, the expression of *ORE1*, *SGR1* and *WRKY70* were similar between Col-0 and the *cipk23* mutant. However, under the 10-0.5 condition, the transcription of *ORE1* and *SGR1* were greatly upregulated, while the transcription of *WRKY70* was inhibited in leaves of the *cipk23* mutant, as compared with Col-0 (Figure 8b). The transcriptional detection of these marker genes after the 16-day treatment showed consistent results with that of the 7-day treatment. Under 10-0.5 condition, differences of the transcriptional level of these genes between Col-0 and the *cipk23* mutant are enlarged (Figure 8d). Therefore, it is speculated that the deletion of *CIPK23* may trigger chlorophyll degradation under high NH_4_^+^/K^+^ ratios.

## 3. Discussion

CIPK23 is a critical regulator in the acquisition of K^+^ and NH_4_^+^ in plants. Under low-K^+^ stress, the protein enhances K^+^ acquisition in roots by the phosphorylation of the K^+^ channel AKT1 and the transporter HAK5 [14,15,17]. Meanwhile, it restrains NH_4_^+^ uptake by inhibiting the activity of ammonium transporter 1 (AMT1, AMT1;1 and AMT1;2) [13]. In this study, the same *Arabidopsis* materials were used, and a comparative analysis between the symptoms and the growth condition was conducted in detail. It was found that the cause of the difference is NH_4_^+^ toxicity triggered by high ratios of NH_4_^+^ to K^+^. To be more precise, the leaf chlorosis phenotype is not the typical symptom of K^+^ deficiency that might be expected; rather, it is a symptom of NH_4_^+^ toxicity, and the sole low-K^+^ condition does not result in leaf chlorosis in *Arabidopsis* (Figure 1, Figure 2 and Figure 4). This is quite consistent with the unexplained phenomenon that the chlorosis phenotype in low-K^+^ condition is also high-NH_4_^+^-dependent [17]. A general problem in the research field of plant potassium nutrition is the difficulty in directly determining whether plants are in a K^+^-deficient state. However, because of the competitive interaction between K^+^ and NH_4_^+^ and the toxic influence of high levels of NH_4_^+^ to plants, the internal environment of low K^+^ concentrations in plant cells or tissues can be indicated by NH_4_^+^ toxicity [1,10,11]. Therefore, a new way to identify plant materials that are sensitive to low-K^+^ or high-NH_4_^+^ conditions has been explored. Further, the study provides a new idea for plant phenotypic identification in which the content of some ions can be evaluated by observing the symptoms triggered by their competitive ions.

Previously, many cultivation experiments have revealed that the occurrence of NH_4_^+^ toxicity always resulted from a decreased uptake of essential cations such as K^+^, Mg^2+^ and Ca^2+^ [21,38]. High levels of NH_4_^+^ suppress the uptake and transport of K^+^ in plants, and high levels of K^+^ also reduce the NH_4_^+^ content [10,21]. However, few experiments have been focused on a detailed elucidation of the correlation of the NH_4_^+^/K^+^ ratio. In our study, Col-0 and the *cipk23* mutant were taken as high-NH_4_^+^ tolerant and sensitive materials, respectively. Then, through a series of gradient NH_4_^+^- and K^+^-supplies tests, it was found that a high NH_4_^+^/K^+^ ratio, but not the absolute high NH_4_^+^ or low K^+^ concentrations, is the main reason for NH_4_^+^ toxicity (Figure 3 and Figure 4). This is a modification to the current understanding of NH_4_^+^ toxicity and is of great importance for fertilization and agricultural production. Currently, because of soil acidification and rain leaching, K^+^ deficiency is a common problem in soils [3]. On the farmland in Asia, free K^+^ is extremely deficient (0.1–1 mM) [3,39,40], while NH_4_^+^ content is much higher (2–20 mM), especially in irrigated paddy fields [2]. The K^+^ situation suggests that it is common for fields to either possess or easily obtain a high NH_4_^+^/K^+^ ratio, although the NH_4_^+^ level in the soils is not so high. NH_4_^+^ toxicity might be a common threat to most crops. Different crops exhibit different tolerances to NH_4_^+^ toxicity [2]. Generally, some industrial crops, including tomato, potato, barley, bean, sugar beet and strawberry, are sensitive to lower NH_4_^+^ levels, while some crops, such as rice, onion and leek, have evolved to be adaptive to high NH_4_^+^ concentrations [2]. Based on our study, the difference in tolerance of these plants might be due to the internal NH_4_^+^/K^+^ ratio that they can endure. The internal balance or homeostasis of NH_4_^+^ and K^+^ in different crops is essential. During agricultural production, it is quite useful and valuable to evaluate the NH_4_^+^/K^+^ ratio of different crops and then make a proper fertilization plan according to their nutrition habits.

Based on the ion content analysis, it was found that in low or high NH_4_^+^/K^+^ ratio conditions, the K^+^ content in wild-type plants was higher than that in the *cipk23* mutant (Figure 1, Figure 2, Appendix A and Figure 6). Under high NH_4_^+^/K^+^ ratio conditions, the NH_4_^+^ content in the *cipk23* mutant was higher than that in wild-type plants. The results were completely consistent with those reported previously [13,14,15,17], which showed that *CIPK23* plays a promotional and inhibitory role in K^+^ and NH_4_^+^, respectively. Meanwhile, *CIPK23* overexpressing lines were also identified to be more tolerant than the control lines in former reports [14,17], with an obvious phenotype of green leaves and longer roots. Here, based on more detailed measurement data in roots, we found that under the high NH_4_^+^/K^+^ ratio solutions, wild-type plants exhibited longer primary roots, lateral roots and secondary lateral roots and more shoot and root biomass, but there was no obvious change in the number of secondary lateral roots (Figure 6 and Appendix A). Confusingly, when seedlings grew in a high NH_4_^+^/K^+^ ratio medium, more lateral roots with longer length were observed in wild-type plants (Figure 6). *CIPK23* greatly promotes the elongation of primary roots, lateral roots and secondary lateral roots and enhances the shoot and root biomass, but its function in the formation of lateral roots needs more evidence. Anyway, all these morphogenetic changes are important characteristics in plant root system architecture, especially in crop cultivation and developmental biology. The underlying mechanism of *CIPK23* in the regulation of root growth and development needs to be elucidated in the future.

## 4. Materials and Methods

### 4.1. Plant Materials, Growth Conditions and Treatment Conditions

*Arabidopsis thaliana* ecotype Columbia was used as the plant material for seedling experiments in this study. The T-DNA insertion mutant *cipk23* (SALK_036154) was ordered from TAIR (https://www.arabidopsis.org), and *ProCIPK23*::*GUS* seeds were provided by Prof. Weihua Wu and Prof. Yi Wang (College of Biological Sciences, China Agricultural University, China). They are all in the Columbia-0 (Col-0) background. Primers of identification and characterization of homozygous *cipk23* T-DNA insertion allele are listed in Appendix A.

For on-plate growth assays, the method of Dong et al. [41] was used. Seeds of *A. thaliana* were surface-sterilized (75% ethanol for 2 min and 100% ethanol for 1 min) and grown in half-strength MS medium containing 3.0% (*w/v*) sucrose and 1.0% (*w/v*) agar (pH 5.8). After 3 days of stratification at 4 °C in the dark, the seeds were positioned vertically under constant illumination at 22 °C. The 5-day-old seedlings (primary root length of approximately 1 cm) were transferred to MS and treatment plates for 7 days. Treatment media were modified according to [17] (Appendix A). KCl was used to change the K^+^ level in the treatment plates. The phenotype pictures were taken by an automatic colony counter (Shineso 2.0). After 7 days of treatment, the roots and shoots were separated carefully. Then, the fresh and dry weights and K^+^ and NH_4_^+^ contents were measured. The number of detections per material under each treatment was approximately 150 plants.

For the hydroponically growth assays, the method of seed germination and culture was modified from previous research [42], as described in Appendix A. Seeds were germinated and seedlings were grown by a hydroponic experimental system (a one-fifth-strength Hoagland Solution, 1/5 HS) using nitrate (NO_3_^−^) as the sole basic nitrogen source. The 1/5 HS of normal culture was supplemented with 1 mM K^+^ with no NH_4_^+^ (0-1) according to Ragel et al. (Appendix A) [15]. Plants were cultured in a growth room (300 μmol photons m^−2^ s^−1^ continuous illumination, 60% relative humidity and 22 °C). The seedlings (15 days) were transplanted in pots (size: 15 × 25 × 40 cm) with 6 L 1/5 HS and grown hydroponically using nutrient medium containing different NH_4_^+^- and K^+^ gradients. In the 1/5 HS, the K^+^ and NH_4_^+^ contents can be freely changed without affecting the other ions. K_2_SO_4_ and NH_4_Cl were added to adjust the levels of K^+^ and NH_4_^+^, respectively. A 10 M NaOH solution was used to adjust the pH (5.6–6.0) of the 1/5 HS solution. The medium was refreshed every 2 days and the positions of the pots were interchanged when refreshing the medium to eliminate edge effects. Plants were grown in each pot, and the growth conditions were the same as described above. After 7 days of treatment, the phenotype pictures were also taken by Shineso 2.0. Then, the fresh and dry weights and K^+^ and NH_4_^+^ contents were measured. The number of detections per material under each treatment was at least 18 plants.

### 4.2. Determination of Shoot and Root Fresh and Dry Weight

The plant samples were washed thoroughly once with 10 mM CaSO_4_ for 5 min and twice with double-distilled water [10]. After six plants, the cleaning solution was renewed. Leaves and roots were carefully detached and collected separately, and the fresh weights (FW) of the different treatments were measured. Fresh shoots and roots were held at 80 °C for 2 days, and then the dry weight (DW) was measured. Then, the samples were milled into fine powders.

### 4.3. Determination of K^+^ and NH_4_^+^ Content

To determine the K^+^ contents, dry samples were digested in 6 mL (approximately 0.2 g DW) of 0.5 M HCl, shaken for 1 h at room temperature, filtered by filter paper into a new tube and measured by a flame photometer (6400A). The readings were used to calculate the K^+^ concentrations in the solution.
K^+^ (mmol g^−1^ DW) = ((A/M) ∗ V ∗ Dilution multiples ∗ 0.001)/m(1)
where A is the concentration calculated by reading according to standard curve (μg mL^−1^); M is the relative molecular mass of K^+^; V is the reading volume (mL); m is the dry weight (g).

To measure the NH_4_^+^ content, the shoots and roots of each sample were washed with 10 mM CaSO_4_ and double-distilled water; the fresh weight was then obtained, and the samples were frozen in liquid nitrogen. To ensure that the samples were well mixed, the shoots and roots of every six plants were mixed with liquid nitrogen, and then extractions were conducted with 6 mL of 10 mM formic acid for the NH_4_^+^ content assay using the *o*-phthalaldehyde (OPA) method, which has been described in a previous report [11]. The sample absorbance was measured at 410 nm by using a spectrophotometer (UV-7502PC, AOE Instruments). The readings were used to calculate the concentrations of the total tissue NH_4_^+^.
NH_4_^+^ (μmol g^−1^ FW) = ((A/M) ∗ V ∗ Dilution multiples)/m(2)
where A is the concentration calculated by reading according to standard curve (μg mL^−1^); M is the relative molecular mass of NH_4_^+^; V is the reading volume; m is fresh weight (g).

### 4.4. Histochemical Staining of Hydrogen Peroxide (H_2_O_2_) and β-Glucuronidase (GUS)

Histochemical staining of H_2_O_2_ was performed as previously described, with minor modifications [43]. The pH of the water was modulated to 3.0. Then, a 0.1 mg mL^−1^ 3,3′-diaminobenzidine (DAB) solution was prepared, and the other steps remained unchanged. A *ProCIPK23*::*GUS* analysis of the plants was performed under multiple growth conditions for 7 days. Shoots and roots of promoter plants were cut off and transferred to tubes (shoot, 50 mL tube; root, 1.5 mL tube) containing a GUS staining solution (LEAGENE Lot.1127A19) and incubated at 37 °C for 24 and 2 h, respectively. The staining solution was then removed and the tissues were stored in 75% ethanol. GUS and H_2_O_2_ staining pictures of shoots were taken using Shineso 2.0. GUS-stained roots were photographed with a LEICA S8APO. Three replicates were completed for all experiments.

### 4.5. RNA Extraction and Real-Time Quantitative PCR (RT-qPCR)

Total RNA of seedlings was extracted with a phenol-based method. Samples were frozen in liquid N_2_ and then ground using mortar and pestle. Homogenized sample was transferred into a 2 mL microcentrifuge tube and mixed with preheated (80 °C) 350 µL phenol water (Solarbio) and 350 µL RNA extraction buffer (100 mM Tris-HCl (pH 8.0), 0.1 M LiCl, 10 mM EDTA (pH 8.0)). The sample was vortexed for 20 s to be sure that the aqueous and organic phases were properly mixed and then placed for 5 min at room temperature. Next, 350 µL chloroform was added, and the sample was vortexed for 20 s before being placed for 5 min at room temperature again. The sample was then centrifuged for 15 min at 12,000 rpm, yielding three main phases. The aqueous phase was transferred to a new RNase-free microcentrifuge tube to which was added 1 volume of 4 M LiCl. This was then vortexed for 20 s and incubated for over 4 h at −20 °C. After incubation, the sample was centrifuged for 15 min at 12,000 rpm at 4 °C. The supernatant was discarded by decanting, and the pellet was washed twice with 70% ethanol. The ethanol was discarded by decanting, traces of ethanol by were eliminated by pipetting and the pellet was air-dried in a hood. The pellet was resuspended in 80 µL RNase-free H_2_O.

To test the expression level of *CIPK23* and some marker genes, the seedlings were treated for 7 days before being used to detect gene expression. cDNA was synthesized from 1 μg total RNA using HiScript III RT SuperMix for qPCR (+gDNA wiper) (Vazyme) for RT-qPCR. qPCRs were performed using ChamQ SYBR qPCR Master Mix (Vazyme) on the LightCycler 96 (Roche). The amplification reactions were performed in a total volume of 20 µL, which contained 10 µL 2× ChamQ SYBR qPCR Master Mix, 7.2 µL ddH_2_O, 0.8 µL forward and reverse primers (10 µM), and 2 µL cDNA (which had been diluted 10 times after synthesis). PCR was conducted as follows: 95 °C for 1 min, followed by 40 cycles of 95 °C for 10 s and 60 °C for 30 min. Primers of qPCRs are listed in Appendix A. All primers referenced previous studies, and expression of *Actin2* was used as an internal control [36,37]. Three biological replicates were included for data quantification.

### 4.6. Quantification and Analyses of Major Fluorescence Parameters of Chlorophyll

To measure the level of etiolation, the chlorophyll contents in the leaves were examined. The chlorophyll contents were measured according to the method described in a previous study [44]. Seedlings which had been treated for 7 days under multiple conditions were weighed and incubated in absolute ethyl alcohol at 4 °C in darkness for 48 h. The absorbance of the resulting extracts was measured at 665 and 649 nm with the spectrophotometer (UV-7502PC, AOE Instruments), and chlorophyll concentration was calculated as milligrams per gram of fresh weight [39].

To determine the photosynthetic performance of the wild type (Col-0) and the *cipk23* mutant, the chlorophyll fluorescence under low NH_4_^+^/K^+^ (0.5 mM NH_4_^+^ and 0.5 mM K^+^) and high NH_4_^+^/K^+^ (10 mM NH_4_^+^ and 0.5 mM K^+^) ratios for 7 days was tested by analyzing the major fluorescence parameters (PSI FluorCam company, FluorCam7 Software), which revealed the PSII activity. The protocol selected was Quenching Act1. The parameters of the software were as follows: UV = 100, EI. Shutter = 1 and Sensitivity = 2.9%.

### 4.7. Measurement of Root Length

Photos of the primary roots of Col-0 and *cipk23* under all conditions for 7 days were taken with a Shineso 2.0. Photos of the first and secondary lateral roots under low NH_4_^+^/K^+^ (0.5 mM NH_4_^+^ and 0.5 mM K^+^) and high NH_4_^+^/K^+^ (10 mM NH_4_^+^ and 0.5 mM K^+^) ratios were taken with a root scanner (Expression 11000XL, ZPSON). Subsequently, the lengths of roots (primary and lateral root) were measured using the ImageJ software (https://imagej.en.softonic.com/).

### 4.8. Statistical and Graphical Analysis

Statistical analysis of the data was developed by using the IBM SPSS Statistics 23 software. Significant differences among treatments were examined by one-way ANOVA using the LSD test at *p* < 0.05. The figures were drawn by GraphPad Prism 6.0. All graphs and images were arranged using Adobe Acrobat XI Pro.

## Figures and Tables

**Figure 1 plants-09-00501-f001:**
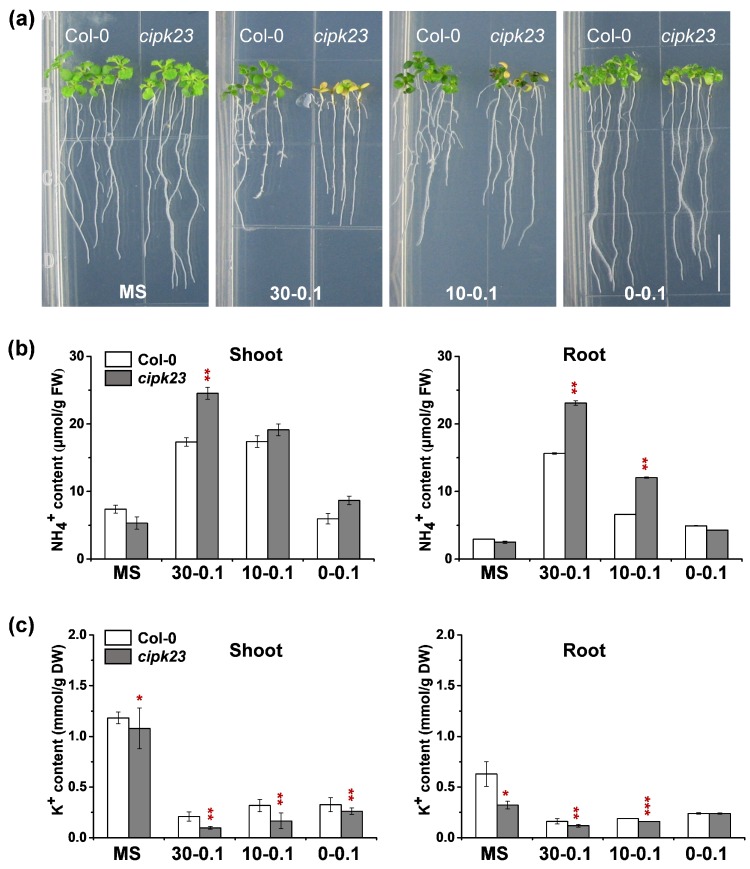
The phenotypic comparison between Col-0 and *cipk23* mutant under different NH_4_^+^ and K^+^ concentrations in medium. (**a**) Phenotype test of 5-day-old seedlings (Col-0 and *cipk23* mutant) grown in MS medium and different concentrations of NH_4_^+^ (30, 10 and 0 mM) and 0.1 mM K^+^ medium. ([NH_4_^+^]-[K^+^]) is used to represent the respective concentrations of NH4^+^ and K^+^ (mM) in each treatment. (*n* = 150 plants). Scale bar = 1 cm. (**b**) NH_4_^+^ content in shoots and roots of *Arabidopsis*. (**c**) K^+^ content in shoots and roots of *Arabidopsis*. *n* = 3 biological replicates. Data are means ± SD. One-way ANOVA with LSD test (* *p* < 0.05, ** *p* < 0.01 and *** *p* < 0.001) was used to analyze statistical significance.

**Figure 2 plants-09-00501-f002:**
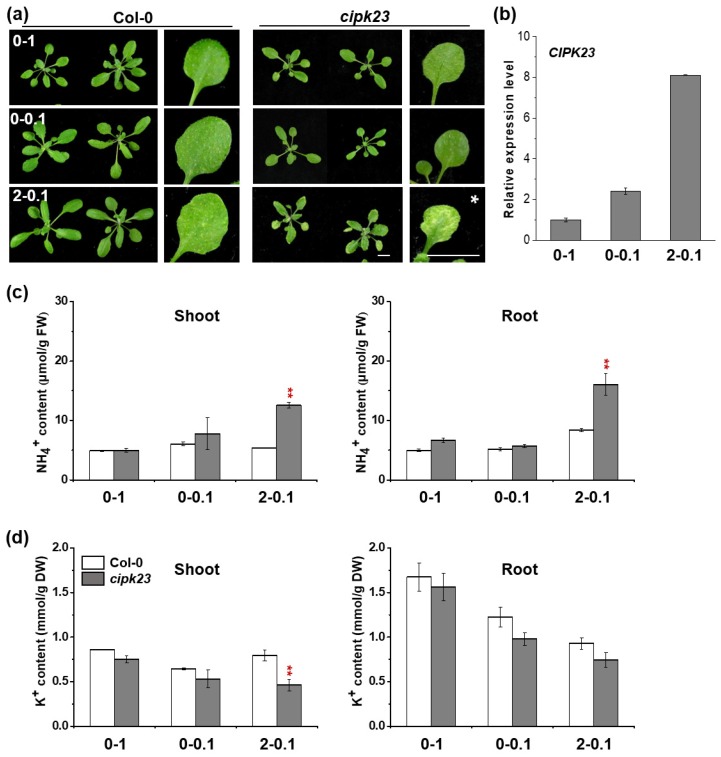
The phenotypic comparison of Col-0 and *cipk23* mutant under different NH_4_^+^ and K^+^ concentrations in hydroponic solution. (**a**) Phenotypic comparison of 15-day-old seedlings (Col-0 and *cipk23* mutant) grown in solutions with different combinations of NH_4_^+^ and K^+^ for 7 days. Scale bar = 1 cm. White stars were used to represent the conditions showing leaf chlorosis phenotype. (*n* = 24 plants). (**b**) RT-qPCR analysis of *CIPK23* expression in shoots. (**c**) NH_4_^+^ content in shoots and roots of *Arabidopsis*. (**d**) K^+^ content in shoots and roots of *Arabidopsis*. *n* = 3 biological replicates. ([NH_4_^+^]-[K^+^]) is used to represent the respective concentrations of NH4^+^ and K^+^ (mM) in each treatment. Data are means ± SD. One-way ANOVA with LSD test (** *p* < 0.01) was used to analyze statistical significance.

**Figure 3 plants-09-00501-f003:**
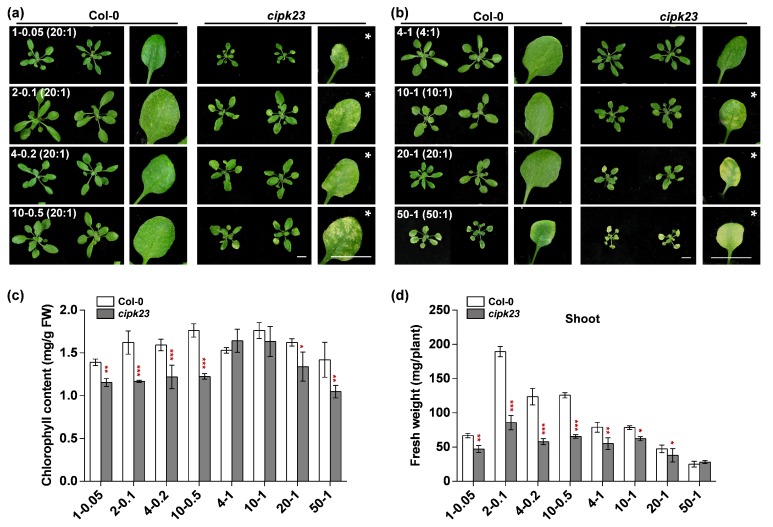
The phenotypic comparison between Col-0 and *cipk23* mutant under high NH_4_^+^/K^+^ ratio conditions. (**a**,**b**) Fifteen-day-old seedlings grown in different NH_4_^+^/K^+^ ratio solutions for 7 days. Scale bar = 1 cm. White stars are used to represent the conditions triggered leaf chlorosis phenotype. (**c**) Chlorophyll contents in shoots of *Arabidopsis* seedlings under different NH_4_^+^/K^+^ ratio treatments. (**d**) Shoot fresh weights of *Arabidopsis* seedlings under different treatments. *n* = 3 biological replicates. ([NH_4_^+^]-[K^+^]) is used to represent the respective concentrations of NH4^+^ and K^+^ (mM) in each treatment. Data are means ± SD. One-way ANOVA with LSD test (* *p* < 0.05, ** *p* < 0.01 and *** *p* < 0.001) was used to analyze statistical significance.

**Figure 4 plants-09-00501-f004:**
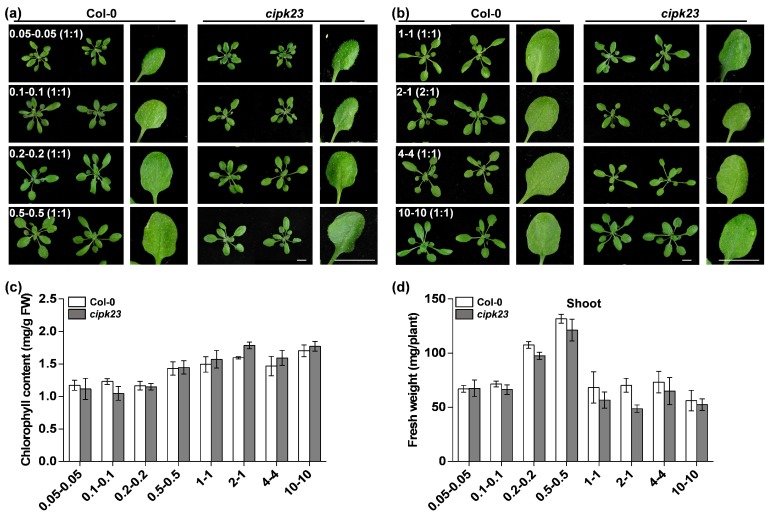
The phenotypic comparison of Col-0 and *cipk23* mutant under low NH_4_^+^/K^+^ ratio conditions. (**a**,**b**) Fifteen-day-old seedlings grown in same NH_4_^+^/K^+^-ratio solutions with different NH_4_^+^ and K^+^ concentrations for 7 days. Scale bar = 1 cm. (**c**) Chlorophyll contents in shoots of *Arabidopsis* seedlings under different treatments. (**d**) Shoot fresh weights of *Arabidopsis* seedlings under different treatments. *n* = 3 biological replicates. Means were separated using ± SD. Variations among treatments was examined by one-way ANOVA using the LSD test.

**Figure 5 plants-09-00501-f005:**
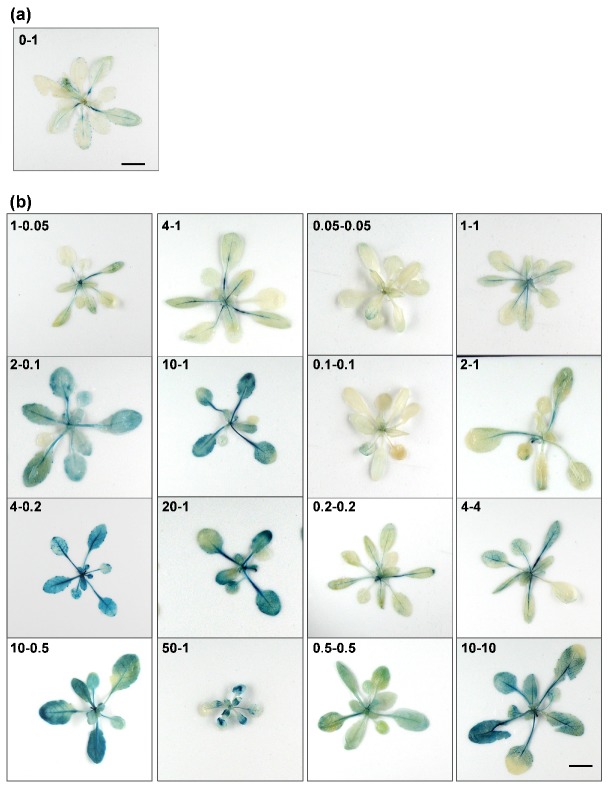
Expression of *CIPK23* under different growth conditions. (**a**,**b**) Fifteen-day-old *ProCIPK23*::*GUS* plants were exposed to various NH_4_^+^/K^+^ ratios for 7 days, and the shoots were collected for GUS staining. (*n* = 3 biological replicates). Scale bar = 1 cm.

**Figure 6 plants-09-00501-f006:**
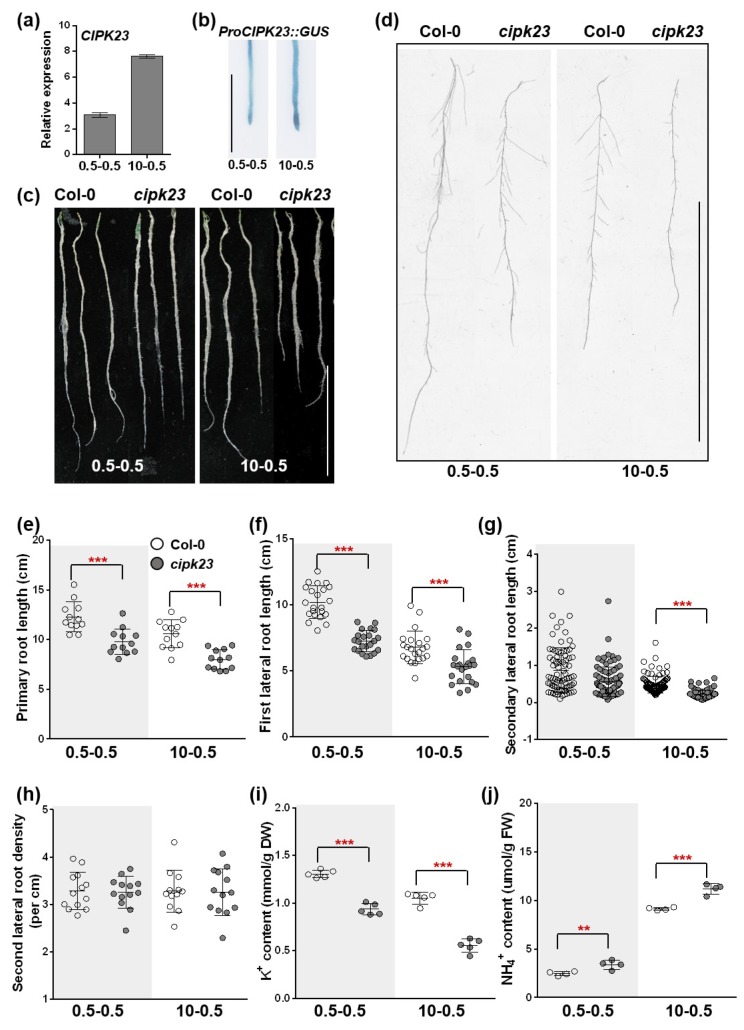
Phenotype and morphogenetic measurement of *Arabidopsis* roots under low NH_4_^+^/K^+^ ratio and high NH_4_^+^/K^+^ ratio treatments. (**a**,**b**) Expression of *CIPK23* in roots under two conditions. Scale bar = 2 mm. (**c**) Roots of Col-0 and *cipk23* mutant seedlings after treatment for 7 days (*n* = 48 plants). Scale bar = 5 cm. (**d**) Representative lateral roots (far from the apical region) after treatment for 7 days (*n* = 12 plants). Scale bar = 5 cm. (**e**–**h**) Morphogenetic measurement of *Arabidopsis* roots under low NH_4_^+^/K^+^ ratio and high NH_4_^+^/K^+^ ratio treatments. (**i,j**) K^+^ content and NH_4_^+^ content of roots after treatment for 7 days (*n* = at least 4 biological replicates). One-way ANOVA with LSD test (** *p* < 0.01 and *** *p* < 0.001) was used to analyze statistical significance.

**Figure 7 plants-09-00501-f007:**
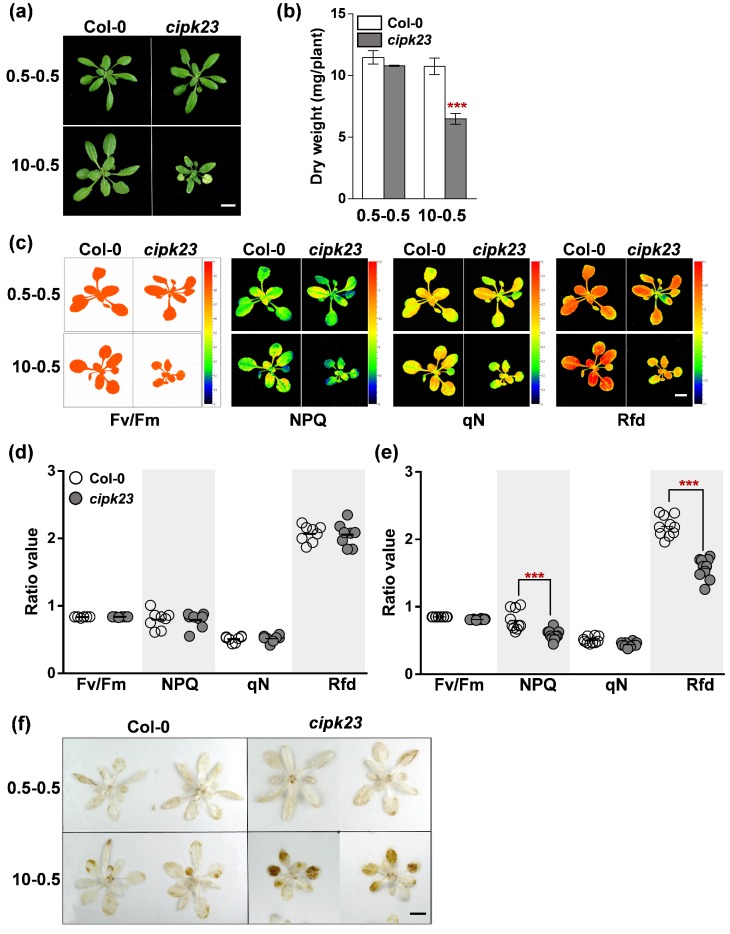
The chlorophyll fluorescence (ChlF) measurements and H_2_O_2_ staining of *Arabidopsis* shoots under low NH_4_^+^/K^+^ ratio and high NH_4_^+^/K^+^ ratio treatments. (**a**) Phenotype comparison between Col-0 and *cipk23* mutant after treatment for 7 days. (**b**) Shoot dry weights of *Arabidopsis* seedlings under 0.5-0.5 and 10-0.5 conditions. (**c**) PSII images captured by FluorCam. Maximum quantum yield of PSII photosystems (F_v_/F_m_), non-photochemical quenching (NPQ and qN) and fluorescence decrease ratio (Rfd) are shown. *n* = 10 plants. (**d**,**e**) Comparison of ChlF parameters of *Arabidopsis* shoots. (**f**) Leaf H_2_O_2_ detection of Col-0 and *cipk23* seedlings (*n* = 9 plants). Scale bar = 1 cm. Data are means ± SD. One-way ANOVA with LSD test (*** *p* < 0.001) was used to analyze statistical significance.

**Figure 8 plants-09-00501-f008:**
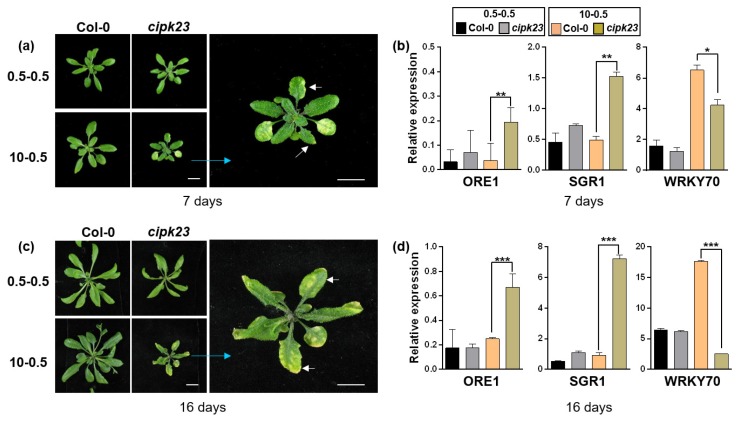
RT-qPCR analysis of stress-related genes. (**a**) Phenotype comparison between Col-0 and *cipk23* mutant after treatment for 7 days. (**b**) Expression of marker genes in leaves after treatment for 7 days. (**c**) Phenotype comparison between Col-0 and *cipk23* mutant after treatment for 16 days. (**d**) Expression of marker genes in leaves after treatment for 16 days. Brilliant blue arrows are used to show enlarged pictures of mutant under 10-0.5 condition. White arrows show the fifth and sixth leaves from the bottom up, which were used for RT-qPCR test in both Col-0 and the *cipk23* mutant. Data are means ± SD. One-way ANOVA with LSD test (* *p* < 0.05, ** *p* < 0.01 and *** *p* < 0.001) was used to analyze statistical significance.

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
