# Peer review of "NH4+ Toxicity, Which Is Mainly Determined by the High NH4+/K+ Ratio, Is Alleviated by CIPK23 in Arabidopsis"

_plants, 2020, doi:10.3390/plants9040501_

Round 1

Reviewer 1 Report

In the research paper "NH4+ Toxicity, Which Is Mainly Determined by High NH4+/K+ Ratio, Is Alleviated by CIPK23 in Arabidopsis” Shi and collaborators tested different concentrations and ratios of NH4+/K+ to understand the NH4+ toxicity effect and the involvement of CIPK23 gene in leaf chlorosis in Arabidopsis, as well as in root development.

The manuscript shows many experimental analyses and high number of results. The experimental design, the objectives and the results are clearly presented. The figures and tables are well presented, as well. The Discussion section is also well written, and the conclusions are supported by the data. The authors also present a promising future work.

Finally, the manuscript can be accepted for publication following only my minor comments highlighted below:

Line 54: remove ‘And’

Line 72: reformulate the sentence ‘What is the essence of leaf chlorosis in cpk23?’

Line 76: replace ‘occupied’ to ‘used’

Line 104: space before ‘The results…

Line 109: ‘Phenotype test’?...

Line 144, 149: be coherent with RT-qPCR…it should be RT-qPCR

Line 198: try not initiate a phrase with ‘And’…

Line 225: space before ‘in roots…’

Line 261: replace ‘demonstrated’ to ‘suggested’

Line 290: ‘… (Figure 8a and 8b), whereas WRKY70…’put comma after parenthesis and continue the phrase

Line 291: ‘ (Figure 8a and 8b)’

Line 292: (Figure 8c and 8d)

Line 293: (Figure 8d)

Line 307: ‘In this study, the same …’

Line 309: ‘in detail’

Line 318-320: the phrase is confusing

Line 338: ‘plants might be due’..

Line 371: ‘Treatment media’…

Line 384: replace ‘adapt’ to ‘adjust’

Line 385: ‘2 days’

Line 424: replace ‘RNA analysis’ to ‘gene expression analysis’

Line 425: describe the heat phenol method or include reference

Line 426: RT-qPCR

Line 432: ‘…of total RNA was performed as described above…’

Line 423-433: the authors should indicate the reference genes used in qPCR data normalization, if actin was used, they should explain whether they have performed a previous experiment to select this gene or they selected it from the literature. Information regarding primers concentrations and cDNA amount used in qPCR reactions are missing. Table S3: the phrase …’an internal standard for normalization of gene expression levels..’ fits more to the Actin gene once the authors did not perform densitometric analysis with the semi-quantitative results but should have done qPCR data normalization with the actin reference gene. The authors should state whether melting curve analysis was performed to confirm specific amplification, or if the PCR products were sequenced. The authors should also indicate the primers efficiency, which should be higher than 90%. Also, the authors should state the software used for primer design, if that was the case.

Author Response

Reviewer’s report: In the research paper "NH4+ Toxicity, Which Is Mainly Determined by High NH4+/K+ Ratio, Is Alleviated by CIPK23 in Arabidopsis” Shi and collaborators tested different concentrations and ratios of NH4+/K+ to understand the NH4+ toxicity effect and the involvement of CIPK23 gene in leaf chlorosis in Arabidopsis, as well as in root development.

The manuscript shows many experimental analyses and high number of results. The experimental design, the objectives and the results are clearly presented. The figures and tables are well presented, as well. The Discussion section is also well written, and the conclusions are supported by the data. The authors also present a promising future work.

Response: We appreciate your constructive and positive comments. We have revised the manuscript carefully based on your comments and the manuscript has been checked by professional English editing service. Each revision was marked in blue in the manuscript. Thank you very much for your time to check the manuscript again!

Minor points:

1) Line 54: remove ‘And’. Line 76: replace ‘occupied’ to ‘used’. Line 104: space before ‘The results…. Line 109: ‘Phenotype test’?... Line 72: reformulate the sentence ‘What is the essence of leaf chlorosis in cpk23?’

Response: Thanks for your suggestion! The manuscript has been revised carefully. All the changes are marked in blue in the manuscript.

2) Line 290: ‘… (Figure 8a and 8b), whereas WRKY70…’put comma after parenthesis and continue the phrase. Line 291: ‘(Figure 8a and 8b)’. Line 292: (Figure 8c and 8d). Line 293: (Figure 8d).

Response: Thanks for your suggestion! The manuscript has been revised carefully. All the changes are marked in blue in the manuscript.

3) Line 318-320: the phrase is confusing.

Response: Thanks for your suggestion! The manuscript has been revised carefully. The changes are marked in blue in the manuscript. Please refer to lines 317-319 on Page 11.

4) Line 371: ‘Treatment media’…. Line 384: replace ‘adapt’ to ‘adjust’. Line 385: ‘2 days’.

Response: We accept your comments. The revised manuscript has been edited carefully. All the changes are marked in blue in the manuscript.

5) Line 424: replace ‘RNA analysis’ to ‘gene expression analysis’.

Line 425: describe the heat phenol method or include reference.

Line 423-433: the authors should indicate the reference genes used in qPCR data normalization, if actin was used, they should explain whether they have performed a previous experiment to select this gene or they selected it from the literature. Information regarding primers concentrations and cDNA amount used in qPCR reactions are missing. Table S3: the phrase …’an internal standard for normalization of gene expression levels.’ fits more to the Actin gene once the authors did not perform densitometric analysis with the semi-quantitative results but should have done qPCR data normalization with the actin reference gene. The authors should state whether melting curve analysis was performed to confirm specific amplification, or if the PCR products were sequenced. The authors should also indicate the primers efficiency, which should be higher than 90%. Also, the authors should state the software used for primer design, if that was the case.

Response: We quite agree with your suggestion.

We replaced ‘RNA analysis’ to ‘gene expression analysis’ and described the heat phenol method for RNA extraction in Material and Methods section. Moreover, we have replaced RT-PCR with real-time quantitative PCR (Figure 8), they showed consistent results. We reformulated 4.5 part in manuscript. Please refer to lines 283-289 on Page 9 and lines 435-458 on Page 14.

Hopefully the current version could meet the requirement for publication.

Reviewer 2 Report

This paper describes the role of the Calcineurin B-like interacting protein kinase 23 (CIPK23) in the response of Arabidopsis to the supply of various amounts of potassium and ammonium. Indeed, it was known that cipk23 mutants were more sensitive to ammonium toxicity since this kinase limits the uptake of ammonium.

It was also known that CIPK23 activity enhances potassium uptake. In this paper the authors suggest that actually what triggers toxicity symptoms in the cipk23 mutants are high NH4+/K+ ratios and not just high concentrations of ammonium ion.

This manuscript is generally well written but the English language could be improved in several sentences. The data and results are in general correctly presented and discussed. Although mainly descriptive, this paper is clearly of interest since it clarifies the cause of ammonium toxicity in mutants of the important CIPK23 kinase that is involved in the regulation of both ammonium and potassium uptake. This paper thus provides a rather complete analysis of the influence of variations in NH4+/K+ ratios in the severity of cipk23 mutants phenotypes and investigates possible explanations for the toxicity symptoms observed in the cipk23 mutant.

Nevertheless, I have several comments which, to my opinion, should be addressed by the authors.

1- One of the main problems with this paper is that only one cipk23 mutant allele was used throughout this study. Therefore, it is difficult to univoqually link the observed phenotypes to the cipk23 mutation.

First the authors need to better describe the properties of the used cipk23 T-DNA mutant (is the mutant null? Was the mutant backcrossed to WT Arabidopsis?). They could also mention that in a previous paper this mutant allele was complemented with the CIPK23 cDNA and complemented mutants were no longer sensitive to ammonium (Straub et al., Plant Cell 2017). This support the findings described in this paper. Moreover other T-DNA alleles seem to be available and should have been used to confirm the observed phenotypes.

2- The expression levels of CIPK23 and of some stress-related genes were measured in roots or in the cpk23 mutant, respectively, in response to different NH4+/K+ ratios. What is the rationale for choosing these genes? In the Material and Methods section it is indicated that the gene expression levels were measured by real-time quantitative PCR while in the Results only semi-quantitative RT-PCR data are shown (Figure 6a and 8). I think that semi-quantitative RT-PCR data are difficult to interpret quantitatively. At least the results of Figure 6a and 8 should be quantified by densitometry and normalized with a constitutively expressed gene (and not ribosomal RNA which is expressed at much higher level) to provide ratios of gene expression between the mutant and the WT control which can be statistically analyzed with at least three biological repeats.

Minor points:

  • Figure 4 could be moved to Supplementary data.
  • Why was the CIPK23-GUS activity not also measured in roots (Figure 5) since CIPK23 expression level was determined in roots (Figure 6a)?
  • Did the in vitro plates with different NH4+/K+ ratios contain nitrate or only ammonium as nitrogen source?

Author Response

Reviewer’s report: This paper describes the role of the Calcineurin B-like interacting protein kinase 23 (CIPK23) in the response of Arabidopsis to the supply of various amounts of potassium and ammonium. Indeed, it was known that cipk23 mutants were more sensitive to ammonium toxicity since this kinase limits the uptake of ammonium.

It was also known that CIPK23 activity enhances potassium uptake. In this paper the authors suggest that actually what triggers toxicity symptoms in the cipk23 mutants are high NH4+/K+ ratios and not just high concentrations of ammonium ion.

This manuscript is generally well written but the English language could be improved in several sentences. The data and results are in general correctly presented and discussed. Although mainly descriptive, this paper is clearly of interest since it clarifies the cause of ammonium toxicity in mutants of the important CIPK23 kinase that is involved in the regulation of both ammonium and potassium uptake. This paper thus provides a rather complete analysis of the influence of variations in NH4+/K+ ratios in the severity of cipk23 mutants phenotypes and investigates possible explanations for the toxicity symptoms observed in the cipk23 mutant.

Response: We are very thankful for the reviewer's evaluation on our work, and appreciate the constructive and useful comments and suggestions. We have revised the manuscript carefully based on your comments and the manuscript has been checked by professional English editing service. Each revision was marked in blue in the manuscript. Hopefully the current version could meet the standard for publication.

Major comments:

1) One of the main problems with this paper is that only one cipk23 mutant allele was used throughout this study. Therefore, it is difficult to univoqually link the observed phenotypes to the cipk23 mutation.

First the authors need to better describe the properties of the used cipk23 T-DNA mutant (is the mutant null? Was the mutant backcrossed to WT Arabidopsis?). They could also mention that in a previous paper this mutant allele was complemented with the CIPK23 cDNA and complemented mutants were no longer sensitive to ammonium (Straub et al., Plant Cell 2017). This support the findings described in this paper. Moreover, other T-DNA alleles seem to be available and should have been used to confirm the observed phenotypes.

Response: We accept your comments.

Even though only one line of mutants was used in our study, we think its reliable to evaluate the function of CIPK23 because this mutant has been widely used to study in many previous researches (Ragel et al., 2015; Straub, Ludewig, & Neuhauser, 2017; Xu et al., 2006). This mutant is indeed the mutant null and the expression level detection of CIPK23 was supplemented in Figure S1. Besides, thanks for your valuable suggestion about complemented mutants, we have pointed out its phenotype in Straub’s paper to support our findings (Straub et al., Plant Cell 2017). Please refer to lines 93-94 on Page 3.

2) The expression levels of CIPK23 and of some stress-related genes were measured in roots or in the cpk23 mutant, respectively, in response to different NH4+/K+ ratios. What is the rationale for choosing these genes? In the Material and Methods section it is indicated that the gene expression levels were measured by real-time quantitative PCR while in the Results only semi-quantitative RT-PCR data are shown (Figure 6a and 8). I think that semi-quantitative RT-PCR data are difficult to interpret quantitatively. At least the results of Figure 6a and 8 should be quantified by densitometry and normalized with a constitutively expressed gene (and not ribosomal RNA which is expressed at much higher level) to provide ratios of gene expression between the mutant and the WT control which can be statistically analyzed with at least three biological repeats.

Response: We quite agree with your suggestion.

(1) We chose these marker genes mainly based on the phenotype of cipk23 under high NH4+/K+ stress, which includes leaf chlorosis (damaged chloroplast) and leaf senescence. In these processes, ORE1 works as a positive regulator in plant senescence while WRKY70 works as a negative regulator. Besides, SGR1 is a critical gene in chlorophyll degradation (Besseau et al., 2012; Jia et al., 2019; Kim et al., 2009; Ren et al., 2009; Sakuraba 2015; Ulker et al., 2007).

(2) We have replaced RT-PCR with real-time quantitative PCR (Figure 8), they showed consistent results. We reformulated 4.5 part in manuscript. Please refer to lines 283-289 on Page 9 and lines 435-458 on Page 14.

Minor points:

1) Figure 4 could be moved to Supplementary data.

Response: Thanks for your suggestion! But we prefer to keep it because Figure 4 indicates quite important information. Combined with Figure 3, Figure 4 showed our point which is only under high NH4+/K+ stress, the cipk23 mutant exhibit chlorosis phenotype.

2) Why was the CIPK23-GUS activity not also measured in roots (Figure 5) since CIPK23 expression level was determined in roots (Figure 6a)?

Response: We are very thankful for the consideration. The CIPK23-GUS activity in roots under every condition was shown in Figure S3, which explained in line 233 on page 8.

3) Did the in vitro plates with different NH4+/K+ ratios contain nitrate or only ammonium as nitrogen source?

Response: We are very thankful for the consideration.

In plates, we used NaNO3 or NH4NO3 as nitrate (NO3-) source (Table S1).

Round 2

Reviewer 2 Report

This revised paper describes the role of the Calcineurin B-like interacting protein kinase 23 (CIPK23) in the response of Arabidopsis to the supply of various amounts of potassium and ammonium. Indeed, it was known that cipk23 mutants were more sensitive to ammonium toxicity since this kinase limits the uptake of ammonium.

This manuscript is generally well written but the English language has been improved.

The authors generally addressed the points I raised in a satisfactory manner. Nevertheless I still have some minor comments about this revised submission. I added these new comments after the authors’ responses.

Major comments:

1) One of the main problems with this paper is that only one cipk23 mutant allele was used throughout this study. Therefore, it is difficult to univoqually link the observed phenotypes to the cipk23 mutation.

First the authors need to better describe the properties of the used cipk23 T-DNA mutant (is the mutant null? Was the mutant backcrossed to WT Arabidopsis?). They could also mention that in a previous paper this mutant allele was complemented with the CIPK23 cDNA and complemented mutants were no longer sensitive to ammonium (Straub et al., Plant Cell 2017). This support the findings described in this paper. Moreover, other T-DNA alleles seem to be available and should have been used to confirm the observed phenotypes.

Response: We accept your comments.

Even though only one line of mutants was used in our study, we think its reliable to evaluate the function of CIPK23 because this mutant has been widely used to study in many previous researches (Ragel et al., 2015; Straub, Ludewig, & Neuhauser, 2017; Xu et al., 2006). This mutant is indeed the mutant null and the expression level detection of CIPK23 was supplemented in Figure S1. Besides, thanks for your valuable suggestion about complemented mutants, we have pointed out its phenotype in Straub’s paper to support our findings (Straub et al., Plant Cell 2017). Please refer to lines 93-94 on Page 3.

Reply to authors response: there is no reference at all to the mutant complementation as described in Straub et al. in lines 93-94? The authors only described the fact that the structure/borders of the T-DNA were verified.

2) The expression levels of CIPK23 and of some stress-related genes were measured in roots or in the cpk23 mutant, respectively, in response to different NH4+/K+ ratios. What is the rationale for choosing these genes? In the Material and Methods section it is indicated that the gene expression levels were measured by real-time quantitative PCR while in the Results only semi-quantitative RT-PCR data are shown (Figure 6a and 8). I think that semi-quantitative RT-PCR data are difficult to interpret quantitatively. At least the results of Figure 6a and 8 should be quantified by densitometry and normalized with a constitutively expressed gene (and not ribosomal RNA which is expressed at much higher level) to provide ratios of gene expression between the mutant and the WT control which can be statistically analyzed with at least three biological repeats.

Response: We quite agree with your suggestion.

(1) We chose these marker genes mainly based on the phenotype of cipk23 under high NH4+/K+ stress, which includes leaf chlorosis (damaged chloroplast) and leaf senescence. In these processes, ORE1 works as a positive regulator in plant senescence while WRKY70 works as a negative regulator. Besides, SGR1 is a critical gene in chlorophyll degradation (Besseau et al., 2012; Jia et al., 2019; Kim et al., 2009; Ren et al., 2009; Sakuraba 2015; Ulker et al., 2007).

(2) We have replaced RT-PCR with real-time quantitative PCR (Figure 8), they showed consistent results. We reformulated 4.5 part in manuscript. Please refer to lines 283-289 on Page 9 and lines 435-458 on Page 14.

Reply to authors response: thank you for using real QPCR for determining gene expression levels. Yet in the Materials and Methods it is mentioned that root RNAs were also extracted but their PCR analysis does not appear in the paper? In Figure 8 b and d, white arrows point to the leaves which were used for RNA extraction and QPCR analysis. What is the rank of these leaves? Were leaves from the same rank used for the Col0 control? The expression levels for ORE1 are difficult to see due to the low expression of this gene. A specific figure for ORE1 with a more appropriate y scale would be more appropriate or alternatively a log scale could be used. A statistical analysis of the significance of the differences in expression levels between the control and the mutant would also be needed.

Author Response

Reviewer’s report: This revised paper describes the role of the Calcineurin B-like interacting protein kinase 23 (CIPK23) in the response of Arabidopsis to the supply of various amounts of potassium and ammonium. Indeed, it was known that cipk23 mutants were more sensitive to ammonium toxicity since this kinase limits the uptake of ammonium.

This manuscript is generally well written but the English language has been improved.

The authors generally addressed the points I raised in a satisfactory manner. Nevertheless, I still have some minor comments about this revised submission. I added these new comments after the authors’ responses.

Response: Thank you so much for carefully reviewing our manuscript again! We appreciate your new comments and we did revisions based on them. We have carefully revised the article. Each revision was marked in blue in the manuscript. Hopefully the current version could meet the standard for publication.

Minor comments:

1) There is no reference at all to the mutant complementation as described in Straub et al. in lines 93-94? The authors only described the fact that the structure/borders of the T-DNA were verified.

Response: We are so sorry that last manuscript we submitted was lack of this important revision. We supplemented the properties of the used cipk23 mutant, and described the information necessary in the new manuscript. Please refer to lines 92-93 of page 2 and lines 94-98 of page 3.

The sentences we supplemented:

The same cipk23 T-DNA insertion mutant (SALK_036154), which was widely used in the previous studies of K+ andNH4+ uptake, was selected. The mutant is verified to be a mutant null and the T-DNA insertion results in a complete loss of CIPK23 transcript (Figure S1). This mutant allele was found to exhibit a severe leaf chlorosis phenotype under low-K+-high-NH4+ stress [17]. When it was complemented with the CIPK23 cDNA, the complemented mutant lines were no longer sensitive to the stress [17]. Meanwhile, the mutant also showed an obvious growth inhibition phenotype to high-NH4+ stress [13].

2) Yet in the Materials and Methods, it is mentioned that root RNAs were also extracted but their PCR analysis does not appear in the paper?

Response: Thanks for your suggestions. The improper description might confuse you. Generally, different RNA samples were extracted, based on different experimental purpose. To investigate the transcriptional changes of CIPK23 in leaf and root tissues under different treatments, RNA was extracted from these tissues after treatments (Figure 2b, 6a). To investigate the transcriptional changes of the marker genes involved in chlorophyll degradation, RNA was extracted from Arabidopsis leaves (Figure 8b, 8d). In order to describe it more clearly, we added some description in the corresponding part.

Please refer to lines 160-161 of page 5, lines 237-238 of page 8, lines 301-305 of page 11 and lines 460-461 of page 15.

3) In Figure 8 b and d, white arrows point to the leaves which were used for RNA extraction and QPCR analysis. What is the rank of these leaves? Were leaves from the same rank used for the Col0 control?

Response: Thanks for your suggestions. White arrows indicate the 5th and 6th leaves (from the bottom up) growing out of the Arabidopsis plants. Generally, based on our stable growth condition, there are always four leaves in each 15-day-old seedlings. And the 5th and 6th leaves are the newly grown leaves after the seedlings were transferred into the treatment solutions. Leaves from the same rank were used for Col-0 control. We supplemented above information in the manuscript. Please refer to lines 301-305 of page 11 and lines 318-319 of page 12.

4) The expression levels for ORE1 are difficult to see due to the low expression of this gene. A specific figure for ORE1 with a more appropriate y scale would be more appropriate or alternatively a log scale could be used. A statistical analysis of the significance of the differences in expression levels between the control and the mutant would also be needed.

Response: Thanks for your constructive suggestions. We made revisions on Figure 8 according to your suggestion. Significance analysis is also supplemented in our data. Please refer to lines 313 of page 12.
